# Healthcare provider-targeted mobile applications to diagnose, screen, or monitor communicable diseases of public health importance in low- and middle-income countries: A systematic review

**Pascal Geldsetzer**[1,2,3☯], **Sergio Flores**[4☯], **Blanca Flores**[5], **Abu Bakarr Rogers**[6], **Andrew Y. Chang**[3,7,8]*

1 Division of Primary Care and Population Health, Department of Medicine, Stanford University; Stanford, California; United States of America, 2 Chan Zuckerberg Biohub; San Francisco, California; United States of America, 3 Center for Innovation in Global Health, Stanford University; Stanford, California; United States of America, 4 Department of Public Health and Caring Sciences, Uppsala University; Sweden, 5 Heidelberg University Hospital, Heidelberg; Germany, 6 Stanford University School of Medicine; Stanford, California; United States of America, 7 Department of Epidemiology and Population Health, Stanford University; Stanford, California; United States of America, 8 Stanford Cardiovascular Institute, Stanford University; Stanford, California; United States of America

☯ These authors contributed equally to this work.
* aychang@stanford.edu

**Data Availability Statement:** All included data were generated from the published literature, with

## Abstract

Communicable diseases remain a leading cause of death and disability in low- and middle-income countries (LMICs). mHealth technologies carry considerable promise for managing these disorders within resource-poor settings, but many existing applications exclusively represent digital versions of existing guidelines or clinical calculators, communication facilitators, or patient self-management tools. We thus systematically searched PubMed, Web of Science, and Cochrane Central for studies published between January 2007 and October 2019 involving technologies that were mobile phone- or tablet-based; able to screen for, diagnose, or monitor a communicable disease of importance in LMICs; and targeted health professionals as primary users. We excluded technologies that digitized existing paper-based tools or facilitated communication (i.e., knowledge-based algorithms). Extracted data included disease category, pathogen type, diagnostic method, intervention purpose, study/target population, sample size, study methodology, development stage, accessory requirement, country of development, operating system, and cost. Given the search timeline, studies involving COVID-19 were not included in the analysis. Of 13,262 studies identified by the screen, 33 met inclusion criteria. 12% were randomized clinical trials (RCTs), with 58% of publications representing technical descriptions. 62% of studies had 100 or fewer subjects. All studied technologies involved diagnosis or screening steps; none addressed the monitoring of infections. 52% focused on priority diseases (HIV, malaria, tuberculosis), but only 12% addressed a neglected tropical disease. Although most reported studies were priced under 20USD at time of publication, two thirds of the records did not yet specify a cost for

individual manuscripts the property of their authors or stakeholders. In accordance with PLOS Digital Health data availability regulations, a full list of analyses which comprise our full dataset is provided in S3 Table with citations, DOIs for all manuscripts, and PMID numbers where available.

**Funding:** The authors received no specific funding for this work.

**Competing interests:** The authors have declared that no competing interests exist.

the study technology. We conclude that there are only a small number of mHealth technologies focusing on innovative methods of screening and diagnosing communicable diseases potentially of use in LMICs. Rigorous RCTs, analyses with large sample size, and technologies assisting in the monitoring of diseases are needed.

## Author summary

Although significant progress has been made in decreasing their worldwide impact, infectious diseases still represent a considerable burden of disease and death. This is especially the case in certain regions of low- and middle-income countries, where limited healthcare resources, personnel, and facilities can make it difficult to provide high quality care. Mobile health (mHealth) technologies are disruptive tools that hold considerable promise in these resource-constrained settings by circumventing some of the aforementioned obstacles. To better understand the availability and characteristics of mHealth technologies for use in low- and middle-income countries, we systematically searched for studies published in English between January 2007 through October 2019 to identify all existing mobile phone- or tablet-based innovations targeted at healthcare providers for use against infectious diseases in these settings and summarized their qualities and performance. We found that four times as many publications focused on tools that simply made data transfer more simple than there were on new tools for detecting or monitoring diseases. Few studies were tested under the most rigorous scientific methods. Many diagnostic technologies we identified require specialized attachments or additional laboratory equipment that connect to the smartphone or tablet, which could make their use in some settings more challenging.

## Introduction

As of 2019, communicable diseases were still the main driver of disability-adjusted life years (DALYs) in children under ten years of age globally and were responsible for six out of the top ten global causes of DALYs [1]. In 2017, 35% of the years of life lost worldwide were from communicable, maternal, neonatal, and nutrition-related disorders [2]. Communicable diseases not only increase mortality and reduce life expectancy in LMICs, but they also cause significant disability, leading to loss of economic productivity in impacted communities [1]. Furthermore, nearly a tenth of the global burden of non-communicable diseases (NCDs) that year were attributed to an infectious cause, with the burden quantified to be 130 million DALYs [3]. Additionally, many LMICs continue to be afflicted by neglected tropical diseases such as dengue virus, Chagas disease, and schistosomiasis. These are not only unique to these regions but also endemic, remaining a major contributor to morbidity and mortality in those settings [4–6].

The persistence of communicable diseases in LMICs is thought to be due to a number of factors, including incomplete development of robust public health infrastructure, shortage of healthcare providers, and continuance of major health disparities [4]. New technologies could help overcome these obstacles to further accelerating the reduction in the communicable disease burden in LMICs. For one, such technologies could enable task shifting from physicians to nurses and community health workers (CHWs) with the goal of alleviating the shortage of more highly trained healthcare worker cadres in low-resource settings. One venue for doing so

involves equipping such personnel with mobile health (mHealth) technologies, whose simplified user interfaces, integrated workflow protocols, and lower costs would be ideal for extending the practice capabilities of their users [7]. For example, incorporating mHealth apps in routine CHW activities has been shown to be beneficial in process improvement and technology development, standards and guidelines, education and training, and leadership and management [8]. mHealth devices have already been demonstrated to improve the management of infectious diseases in many instances in low-resource settings, [9–11] as they can serve as rapid and cheap diagnostic tools [12,13]. The wireless, portable aspects of many such technologies also increase the accessibility of healthcare services to patients by reducing travel time and expenses [8].

The current published literature contains many reports of applications that digitize existing knowledge-based algorithms or facilitate inter-provider or patient-provider communication. To the best of our knowledge, however, it does not offer a comprehensive, up-to-date systematic review of truly innovative, novel provider-facing mHealth technologies available for infectious disease care in LMIC settings. These include technologies such as simplified laboratory testing equipment with smart device interfaces and artificial intelligence-guided diagnostic tools. As such, we conducted a systematic review that aims to identify all existing novel mobile phone- or tablet-based innovations targeted at healthcare providers and summarize the performance of these technologies.

## Methods

### Inclusion and exclusion criteria

We searched the literature and screened titles and abstracts of articles (and if inconclusive, the full-text versions of articles) using the following inclusion criteria:

i. The technology reported must be mobile phone or tablet-based for their clinical function—this excludes mobile devices and applications that solely use their internet connectivity to transmit data;

ii. The technology must target healthcare professionals specifically as users—tools used to educate patients, change patient behaviors as consumer products, or improve patient-provider communication were excluded;

iii. The technology reported must be able to screen, diagnose, or monitor a disease;

iv. The technology must represent an innovation—applications solely used to keep records, reproduce existing guidelines and clinical calculators in digital form, or facilitate communication between providers, or digitizing knowledge-based algorithms were excluded [14]. This criterion was no present in in our prespecified inclusion criteria outlines in the protocol, and was added during the screening process based on emerging patterns and the need to focus on innovative solutions for disease screening, diagnosis, and monitoring in LMICs;

v. The disease the technology is designed to address must be a communicable disease of public health importance for LMICs. Such diseases were defined as infectious conditions that are estimated to cause more than 1% of deaths in any five-year age group in the general population or among neonates, or infectious diseases that have a prevalence of more than 0.1% in any five-year age group in the general population or among neonates. The Global Burden of Disease Project's 2019 estimates were utilized for this appraisal [15].

vi. The articles should be published in English with full text available and not fall under the category of systematic reviews or study protocols.

Of note, given the timeline of the search, studies involving Coronavirus Disease 2019 (COVID-19) were not included in the present analysis.

## Search strategy

We searched for all studies published in English from January 2007 through October 2019 in the following databases: Cochrane Central (searched on September 30th, 2019), PubMed (searched on October 7th, 2019), and Web of Science (searched on October 7th, 2019). The databases were queried using keywords and medical subject headings (MeSH) combining three major search concepts: namely, the concepts of "mobile/tablet", AND "application/software" AND "diagnostics/monitoring". Specific terms included those attributable to smartphones, tablets, mobile applications, diagnosis, screening, and monitoring. A full list of the search terms used for each database are shown in **S1 Table**. The database searches, examination of abstracts, and inspection of articles' full-text versions were not conducted in duplicate. No restrictions were placed on study design, sample size, or publication type. Finally, the reference lists of all included studies, relevant review articles, and commentaries were screened for additional references. The search process is summarized in **Fig 1**. The review was registered in The International Prospective Register of Systematic Reviews (PROSPERO; Registration number: CRD42020193945) [16]. Of note, the protocol was amended following preliminary screening to narrow the search to communicable disease, to target healthcare providers, and only focus on innovative technologies. These changes were necessitated due to the infeasibly broad scope of the original question of mHealth in LMICs. As such, the entire screening process was rerun *de novo* following the protocol change. Ethical approval was not sought from the Stanford institutional review board as the study did not constitute human subjects research and consisted only of meta-research (which is exempt by definition).

## Data extraction

The following data were extracted from each included article: author(s), title, disease or risk factor, clinical domain by MeSH [17], intervention name, intervention type, purpose and aim of the intervention, target population, type of diagnostic method, type of pathogen studied (by microbial class, LMIC priority disease (namely Human Immunodeficiency Virus (HIV), tuberculosis and malaria), as well as neglected tropical disease (NTD) status as defined by the World Health Organization [18], type of mobile device utilized, type of software, operating system used by intervention, study population and sample size, study methods, stage of development, cost in US dollars (USD) at the time of publication (all dollar figures are given as published in the manuscript and not adjusted for inflation, and in the case of articles reporting currencies other than dollars, were converted to 2021 US dollars [19]), country of development based on first authors' institutional affiliations, location of testing based on the study population country of residence, institutional nation of all listed authors, year of publication, and a summary of the tool (**S2 Table**). These data were extracted qualitatively using Microsoft Excel (Redmond, WA).

## Data analysis

Quantitative data were summarized with counts and proportions. The retrieved data were organized into three themes: epidemiology, technology, and methodology. The epidemiology theme described the disease of interest (and whether it is categorized as an LMIC priority

disease by the Global Health National Academies of Science [20] or diseases that were among the top ten in terms of disability-adjusted life years caused globally in 2019 [21]), its characteristics, and the geographic location of the intervention's development. The technology theme described the primary hardware platform of the innovation, necessary peripherals, as well as the operating system it utilized and its cost considerations. The methodology theme evaluated the phase of study and research design of each publication. **S2 Table** lists these themes, as well as the categories, subcategories, and definitions that accompany them. To elucidate trends among the studies, we created tables that crossed clinical categories and included all the subthemes. We decided against conducting a meta-analysis due to the substantial degree of heterogeneity in study designs, outcome measurements, and reporting of results. As such, we employed a qualitative measure of study quality on a three-tiered (-, +, and ++) system to characterize publication quality as unsound, suboptimal, or sound, based upon the *British Medical Journal*'s Evidence Based Medicine Best Practice Toolkit [22].

## Results

Our initial search of all the above databases retrieved 13,262 results. After duplicates were removed, abstracts screened, full texts reviewed, and articles identified from reference lists of included articles were added, 33 studies met our inclusion criteria (**Fig 1, S3 Table**). Articles were excluded if they described or evaluated: i) non-mobile technology-based interventions (n = 85); ii) interventions targeting patients instead of health professionals as users (n = 43); iii) interventions not meant for diagnosis, screening and/or monitoring (n = 49); iv) interventions adapting extant/current technologies (n = 48); v) presented technology that digitalized knowledge based algorithms that could be done on paper (n = 103); vi) noncommunicable diseases (n = 317), or that vii) did not have a full text available (n = 41); viii) were not available in English (n = 14); ix) were systematic reviews (n = 36); or x) were study protocols or involved non-human testing (n = 8). An overview of the included studies' characteristics is presented in **Table 1** and the full list of identified studies is available in **S3 Table**.

### Epidemiology

Most studies described technologies tested predominantly in the United States (13/33), the rest of the Americas (3/33), followed by Africa (9/33), then Asian countries (5/33) and Europe (3/33). The affiliation of the first author's institutions is located predominantly in the United States (24/61), with fewer based in the Americas (7/61), Europe (14/61), Africa (9/61) and Asia (7/61). A noteworthy observation is that all the studies except one (32/33) involved at least one researcher affiliated with a high-income country institution, even if the research was ultimately conducted in an LMIC.

Most of the identified technologies focus on the diagnosis of communicable diseases (26/35), while the rest aim to screen for (9/35) these diseases. No study expressed monitoring as the main aim of their technology. The diagnostic method of choice chosen by the researchers was most often serological methods (11/35), followed by direct visualization of the microorganisms (10/35) and nucleic acid detection (10/35) (**Fig 2**). Two manuscripts examined machine learning/artificial intelligence-based innovations. The technologies targeted viral (18/42), bacterial (14/42), and parasitic (10/42) infections.

Almost half (17/33) of the included studies addressed an LMIC priority disease. Only a small number of technologies (4/33) targeted a neglected tropical disease. **Table 2** describes the 16 studies of technologies aimed at diseases that were among the top ten in terms of disability-adjusted life years (DALYs) caused globally in 2019. Specifically, these studies targeted drug-susceptible tuberculosis, malaria, diarrheal diseases, and lower respiratory infections.

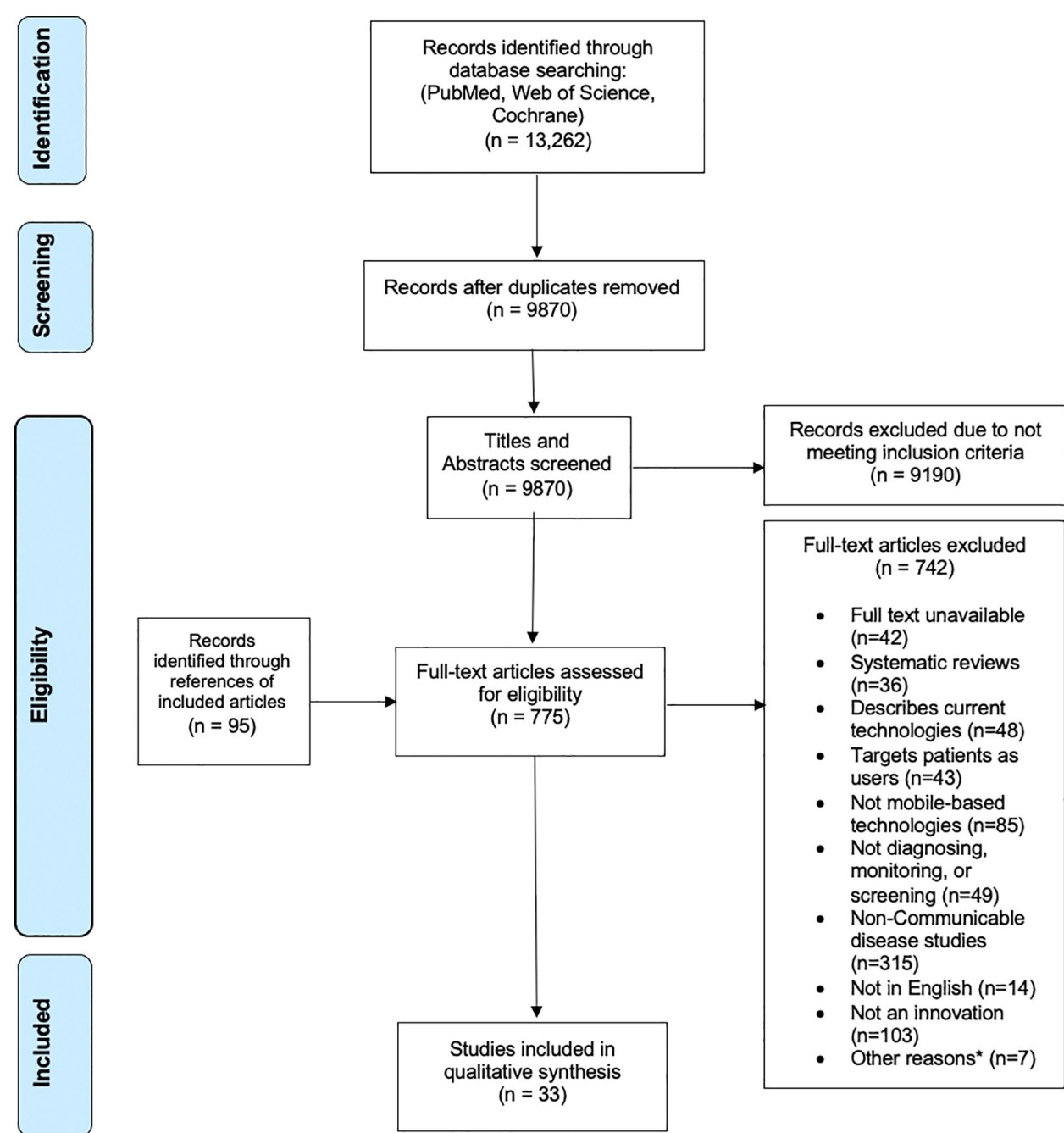

*Other reasons: abstracts, protocols, personal reviews, nonhuman testing.*
From: Moher D, Liberati A, Tetzlaff J, Altman DG, The PRISMA Group (2009). *Preferred Reporting Items for Systematic Reviews and Meta-Analyses: The PRISMA Statement. PLoS Med 6(7): e1000097. doi:10.1371/journal. pmed1000097*

For more information, visit www.prisma-statement.org.

**Fig 1. This figure presents the systematic review flow PRISMA diagram of the screening and exclusion process for articles identified and ultimately included in the analysis.**

**Table 1. Characteristics of Studies.**

| | | Count | Percentage of Total |
|---|---|---|---|
| Year of Publication (Total N = 33) | 2006–2008 | 0 | 0% |
| | 2009–2011 | 1 | 3% |
| | 2012–2014 | 6 | 18% |
| | 2015–2017 | 15 | 45% |
| | 2018–2020 | 11 | 33% |
| Location of Study (Total N = 33) | United States | 13 | 39% |
| | Americas (excluding the United States) | 3 | 9% |
| | Europe | 3 | 9% |
| | Africa | 9 | 27% |
| | Asia | 5 | 15% |
| Affiliation of Researchers (Total N = 61) | United States | 24 | 39% |
| | Americas (excluding the United States) | 7 | 11% |
| | Europe | 14 | 23% |
| | Africa | 9 | 15% |
| | Asia | 7 | 11% |
| Aim (Total N = 35) | Diagnose | 26 | 74% |
| | Screen | 9 | 26% |
| | Monitor | 0 | 0% |
| Diagnostic Method (Total N = 35) | Direct Visualization | 10 | 29% |
| | Serology | 11 | 31% |
| | Antigen Detection | 1 | 3% |
| | Nucleic Acid Detection | 10 | 29% |
| | Others | 3 | 9% |
| Type of Pathogen Studied (Total N = 42) | Viral | 18 | 43% |
| | Bacterial | 14 | 33% |
| | Parasitic | 10 | 24% |
| Type of Device (Total N = 36) | Armband/ Smartwatch | 0 | 0% |
| | Smartphone | 29 | 81% |
| | Non-Smartphone Mobile Phone | 1 | 3% |
| | Tablet | 4 | 11% |
| | iPod Device | 1 | 3% |
| | Another wireless device | 1 | 3% |
| Requires Use of Accessories (Total N = 33) | Yes | 26 | 79% |
| | No | 7 | 21% |
| Development Stage (Total N = 33) | Proof of Concept/Principle | 1 | 3% |
| | In development | 1 | 3% |
| | Prototype | 11 | 33% |
| | Pilot | 0 | 0% |
| | Validation Trial/Test in Clinical Trial | 2 | 6% |
| | Available/Developed | 17 | 52% |
| | Not specified | 1 | 3% |
| Operating System (Total N = 34) | iOS | 11 | 32% |

*(Continued)*

**Table 1.** (Continued)

|  |  | Count | Percentage of Total |
|---|---|---|---|
|  | Android | 18 | 53% |
|  | Windows | 1 | 3% |
|  | Not specified | 4 | 12% |
| Cost at Time of Publication (Total N = 33) | 0–20 USD | 8 | 24% |
|  | 21–100 USD | 2 | 6% |
|  | Over 100 USD | 1 | 3% |
|  | Not specified/no costing yet | 22 | 67% |
| Study Population Sample Size (Total N = 33) | 1–30 | 7 | 21% |
|  | 31–100 | 6 | 18% |
|  | 101–500 | 4 | 12% |
|  | 501–1000 | 1 | 3% |
|  | >1000 | 3 | 9% |
|  | None/Not specified | 12 | 36% |
| Study Design (Total N = 33) | Randomized Clinical Trials | 4 | 12% |
|  | Observational Cohort Studies / Case-Control Studies | 9 | 27% |
|  | Qualitative Studies | 1 | 3% |
|  | Product / Technical Description | 19 | 58% |
| Study Quality (Total N = 33) | - (unsound) | 0 | 0% |
|  | + (suboptimal) | 8 | 24% |
|  | ++ (sound) | 25 | 76% |
| Evaluation Values Used (Total N = 39) | Measures of Diagnostic Accuracy | 22 | 56% |
|  | Variability Measures | 5 | 13% |
|  | Correlation Values | 3 | 8% |
|  | Intraobserver and interobserver values | 1 | 3% |
|  | Measurement Error Analysis | 0 | 0% |
|  | Diverse Measurement Results | 5 | 13% |
|  | Bland Altman Analysis | 1 | 3% |
|  | None/Not specified | 2 | 5% |

## Technology

The most popular device used in the studies was the smartphone (29/36), followed by tablets (4/36) and mobile phones without smartphone capabilities (1/36). Technologies were predominantly developed for the Android operating system (18/34) and Apple iOS (iPhone Operating System) operating system (11/34), with Windows use present in just one product (1/34). Four publications did not specify an operating system used by their application. Most (26/33) of the mobile technologies required the use of peripheral accessories attached to them such as additional optical components, 3D printed attachments, foldscopes, cradles, and dongles. Cost data were not available for most (22/33) technologies. For the technologies with costing information (11/33), most were priced at less than 20 USD (8/11), followed by between 20 and 100 USD (2/11) and one over 100 USD (1/11) at the time of study publication.

## Methodology

Most studies focused on technologies in an advanced development stage, i.e., already developed and/or commercially available (17/33) followed by studies describing prototypes (11/33). Regarding research design, most studies focused on descriptions of the technology without a formal evaluation of its efficacy or effectiveness (19/33) or assessed the technology using an

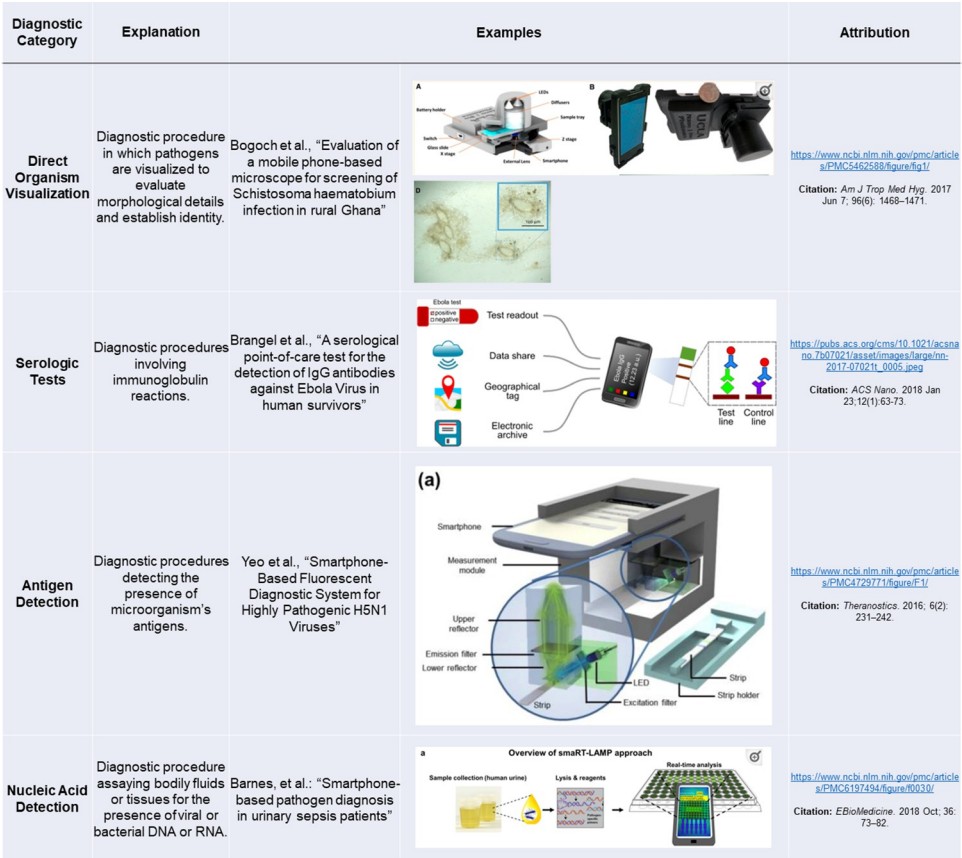

| Diagnostic Category | Explanation | Examples | Attribution |
|---|---|---|---|
| **Direct Organism Visualization** | Diagnostic procedure in which pathogens are visualized to evaluate morphological details and establish identity. | Bogoch et al., "Evaluation of a mobile phone-based microscope for screening of Schistosoma haematobium infection in rural Ghana" | https://www.ncbi.nlm.nih.gov/pmc/articles/PMC5462588/figure/fig1/ Citation: *Am J Trop Med Hyg.* 2017 Jun 7; 96(6): 1468–1471. |
| **Serologic Tests** | Diagnostic procedures involving immunoglobulin reactions. | Brangel et al., "A serological point-of-care test for the detection of IgG antibodies against Ebola Virus in human survivors" | https://pubs.acs.org/cms/10.1021/acsnano.7b07021/asset/images/large/nn-2017-07021t_0005.jpeg Citation: *ACS Nano.* 2018 Jan 23;12(1):63-73. |
| **Antigen Detection** | Diagnostic procedures detecting the presence of microorganism's antigens. | Yeo et al., "Smartphone-Based Fluorescent Diagnostic System for Highly Pathogenic H5N1 Viruses" | https://www.ncbi.nlm.nih.gov/pmc/articles/PMC4729771/figure/F1/ Citation: *Theranostics.* 2016; 6(2): 231–242. |
| **Nucleic Acid Detection** | Diagnostic procedure assaying bodily fluids or tissues for the presence of viral or bacterial DNA or RNA. | Barnes, et al.: "Smartphone-based pathogen diagnosis in urinary sepsis patients" | https://www.ncbi.nlm.nih.gov/pmc/articles/PMC6197494/figure/f0030/ Citation: *EBioMedicine.* 2018 Oct; 36: 73–82. |

*Innovation schematics are reproduced from the cited publications and remain the property of their respective authors/publishers. Attribution links are provided for the image sources and licenses.*

**Fig 2. This figure visually presents several examples of mHealth technologies identified in our screen by functionality.** Explicit written permission to reproduce innovation schematics was obtained from source manuscript corresponding authors.

observational cohort design (9/33). Only a few technologies were evaluated using a randomized design (4/33). Most publications reported study population sizes of less than 30 participants (7/33), followed by study sizes between 31 and 100 participants (6/33), then by study sizes between 101 and 500 (4/33) and over 1,000 subjects (1/33). Twelve studies did not specify a study population size.

## Discussion

### Principal results

The aim of our study was to identify and describe mobile-based technologies targeted specifically at healthcare workers to screen, diagnose, and monitor communicable diseases of public health importance in LMICs. We focused on technologies that constituted a new tool rather than digitizing an existing paper-based tool (i.e., knowledge-based algorithm) or providing a means of communicating between healthcare providers. Our screening found that there were four to five times as many publications on tools that facilitated communication, transferred data, or digitized an existing paper-based algorithm than there were on truly new tools for screening, diagnosing, and monitoring diseases. Additionally, we found that most technologies described in our study were tested in high-income countries using predominantly smartphones as mobile device and Android as the operating system of choice. All but one of the

**Table 2. Studies of technologies addressing diseases among the top ten in disability-adjusted life years globally in 2019.**

| Title | Authors | Disease/ Risk factor | Pathogen Name | Pathogen Family/ Category | Mobile Device Type | Operating System | Diagnostic Method | Clinical Domain | Researchers' Country (or countries) | Country where Research was Conducted |
|---|---|---|---|---|---|---|---|---|---|---|
| App-based symptoms screening with Xpert MTB/RIF Ultra assay used for active tuberculosis detection in migrants at point of arrivals in Italy: The E-DETECT TB intervention analysis. | Barcellini, L. et al. | Pulmonary tuberculosis | *Mycobacterium tuberculosis* | Mycobacteriaceae/ Opportunistic infection | Smartphone | Android | Nucleic Acid Detection/ Organism visualization | Infectious diseases specialists/ Pulmonology | Italy/United Kingdom | Italy |
| Evaluation of a Mobile Phone-Based Microscope for Screening of Schistosoma haematobium Infection in Rural Ghana. | Bogoch, I. et al. | Schistosomiasis | Schistosoma haematobium | Soil-transmitted helminthiasis | Smartphone | Windows | Organism visualization | Infectious diseases specialists/ Pediatrics/ Family medicine | United States/ Canada/ Ghana | Ghana |
| Mobile phone based clinical microscopy for global health applications. | Breslauer, D. et al. | Malaria/ Pulmonary TB | P. falciparum/ M. tuberculosis | Vector Borne Diseases/ Mycobacteriaceae-Opportunistic infection | Mobile Phone | Symbian | Organism visualization | Infectious diseases specialists | United States | United States |
| Evaluation of Malaria Diagnoses Using a Handheld Light Microscope in a Community-Based Setting in Rural Cote d'Ivoire. | Coulibaly, J. et al. | Malaria | Plasmodium falciparum | Vector Borne Diseases | Smartphone | iOS | Organism visualization | Infectious diseases specialists | Côte d'Ivoire/ Switzerland/ United States/ Canada | Côte d'Ivoire |
| Diagnosis of Schistosoma haematobium infection with a mobile phone-mounted Foldscope and a reversed-lens CellScope in Ghana. | Ephraim, R. et al. | Schistosomiasis | Schistosoma haematobium | Soil-transmitted helminthiasis | Smartphone | iOS | Organism visualization | Infectious diseases specialists/ Pediatrics/ Family medicine | United States/ Canada/ Ghana/ Switzerland | Ghana |
| mPneumonia: Development of an Innovative mHealth Application for Diagnosing and Treating Childhood Pneumonia and Other Childhood Illnesses in Low-Resource Settings. | Ginsburg, A. et al. | Pneumonia | Not specified | Not specified | Tablet | Android | Mobile health (mHealth)-based applications (Integrated Management of Childhood Illness algorithm) | Infectious diseases specialists/ Pediatrics/ Family medicine | United States/ Ghana | Ghana |

*(Continued)*

**Table 2.** (Continued)

| Title | Authors | Disease/ Risk factor | Pathogen Name | Pathogen Family/ Category | Mobile Device Type | Operating System | Diagnostic Method | Clinical Domain | Researchers' Country (or countries) | Country where Research was Conducted |
|---|---|---|---|---|---|---|---|---|---|---|
| A point-of-need enzyme linked aptamer assay for Mycobacterium tuberculosis detection using a smartphone | L. Li, Z. Liu, H. Zhang et al | Pulmonary tuberculosis | Mycobacterium tuberculosis | Mycobacteriaceae/ Opportunistic infection | Smartphone | Android | Nucleic acid detection | Infectious diseases specialists/ Pulmonology | China | China |
| Rapid electrochemical detection on a mobile phone | Lillehoj, Peter B.; Ming-Chun Huang et al | Malaria | *Plasmodium falciparum* | Vector Borne Diseases | Smartphone | Android | Nucleic acid detection | Infectious diseases specialists | United States | United States |
| Integrated rapid-diagnostic-test reader platform on a cellphone | Mudanyali, Onur; Stoyan Dimitrov, Uzair Sikora, et al | Malaria/ TB/ HIV | P. falciparum, P. vivax, P. ovale and P. malariae/ M. tuberculosis/ HIV | Vector Borne Diseases/ Mycobacteriaceae-Opportunistic infection/ STD | Smartphone | Android and iOS | Serology | Infectious diseases specialists/ Internal Medicine | United States | United States |
| Mobile phone-based evaluation of latent tuberculosis infection: proof of concept for an integrated image capture and analysis system | Naraghi, Safa; Tinashe Mutsvangwa, René Goliath et al | Latent TB | Mycobacterium sp | Mycobacteriaceae/ Opportunistic infection | Smartphone | Android | Tuberculin skin test induration. | Infectious diseases specialists/ Internal Medicine | South Africa/ United Kingdom | South Africa |
| The Malaria System MicroApp: A New, Mobile Device-Based Tool for Malaria Diagnosis | Oliveira; Allisson Dantas, Clara Prats, Mateu Espasa, et al | Malaria | Plasmodium falciparum | Vector Borne Diseases | Tablet | Android | Organism visualization | Infectious diseases specialists | Brazil/ Spain | Brazil |
| Malaria Diagnosis Using a Mobile Phone Polarized Microscope | Pirnstill, C.W. & Coté, G.L. | Malaria | Plasmodium chabaudi | Vector Borne Diseases | Smartphone | iOS | Organism visualization | Infectious diseases specialists | United States | United States |
| Artificial neural network models to support the diagnosis of pleural tuberculosis in adult patients | Seixas, J.M. et al. | Pleural Tb | Mycobacterium tuberculosis | Mycobacteriaceae/ Opportunistic infection | Tablets | Not specified | Artificial neural net-works (ANN) | Infectious diseases specialists | Brazil/ Canada | Brazil |
| A paper-based microfluidic Dot-ELISA system with smartphone for the detection of influenza A | Wu, Di et al | Influenza | Influenza A virus | Orthomyxoviridae | Smartphone | Android | Serology | Infectious diseases specialists/ Family medicine | United States/ China | China |

*(Continued)*

**Table 2.** (Continued)

| Title | Authors | Disease/ Risk factor | Pathogen Name | Pathogen Family/ Category | Mobile Device Type | Operating System | Diagnostic Method | Clinical Domain | Researchers' Country (or countries) | Country where Research was Conducted |
|---|---|---|---|---|---|---|---|---|---|---|
| Deep Learning for Smartphone-based Malaria Parasite Detection in Thick Blood Smears | Yang, Feng et al | Malaria | Plasmodium falciparum | Vector Borne Diseases | Smartphone | Android | Organism visualization | Infectious diseases specialists | Unites States/ China/ Thailand | Bangladesh/ Thailand |
| Smartphone-Based Fluorescent Diagnostic System for Highly Pathogenic H5N1 Viruses | Yeo, Seon-Ju et al | Avian influenza | H5N1 virus | Orthomyxoviridae | Smartphone | Android | Serology | Infectious diseases specialists/ Family medicine | Republic of Korea/ Vietnam/ United States | Vietnam/ Republic of Korea |

included studies involved at least one author affiliated with a high-income country research institution, with 42% of first authors reporting institutional affiliations in the United States or Europe.

Although half of the technologies were already at an advanced stage of development, few were tested under the rigor of large-scale randomized controlled studies. In general, the sample size was small, with 62% of the studies reporting 100 or fewer subjects. Over half of the included publications were simply technical descriptions of a product. Though most reported studies are of relatively affordable innovations (most under 20 USD), two thirds of the records did not yet specify a price point for the study technology. Most importantly, all the technologies were involved in diagnosis or screening for diseases—none were found to address monitoring of infections. We were, however, encouraged to note that half of the identified technologies focused on LMIC priority communicable diseases such as HIV, malaria and tuberculosis, although only 12% addressed a neglected tropical disease.

Controlling communicable disorders requires prompt screening, diagnosis, and monitoring of the infectious agent, both to treat the disease in the individual and to prevent its further transmission. A plethora of diagnostic tests and procedures have been available to the medical community for decades, and yet, LMICs are still burdened with high levels of communicable diseases [23]. This has been partly explained by poor availability of timely, high-quality diagnostic testing. Diagnostic laboratories in LMICs are usually poorly equipped or sparsely distributed [24], limiting their ability to provide accurate and rapid information to clinicians [25]. Furthermore, the costs of building and maintaining laboratories tends to be prohibitive in resource-constrained settings [24], and training specialized technical personnel requires further financial and logistic investments that are often unavailable in these countries. Our findings seem to suggest that efforts in the development of mobile technologies have also identified laboratory- and imaging-based testing as key obstacles, with approximately four out of five of our included studies focusing on diagnosis rather than screening or monitoring.

Furthermore, many of the diagnostic technologies we identified require the use of structural appendices, optical components, or specialized laboratory equipment that connect to the smartphone/tablet and its inherent software and hardware specifications. Therefore, these devices are not intended to completely replace standard diagnostic/screening tests and procedures, but rather to make them more accessible to professionals in resource-constrained settings. We note their importance here over standalone point of care diagnostic devices that do not interface with mobile devices, as the former facilitate transfer of the diagnostic attachment's results between users who may not share the device and allows for the manipulation of the results within the mHealth environment. That said, these innovations are also limited by their disproportionate reliance on Apple iOS operating systems (as LMIC mobile devices tend to run on Google Android operating systems [26,27]) and frequent lack of large-scale rigorous evaluation in LMIC settings [26].

The rather small number of innovations in this sphere reflects the likely limited public health impact of the presently available device marketplace. Nevertheless, testing of these technologies in LMICs, the wide range of diagnostic methods employed, and the approach to a variety of emerging infectious pathogens that are being diagnosed using these devices are encouraging findings. These would seem to indicate that not only are these technologies being developed, but some are also entering a diversification phase, which may hold promise for the field [28]. Such general findings are consistent with similar work focusing on the mHealth innovations available for use to managed noncommunicable diseases in LMICs [11]. Future work by mHealth researchers could focus on technologies that can be scaled in a way that allows for widespread and cost-effective implementation in resource-constrained health

systems, while also expanding their use to screen and monitor diseases rather than solely diagnose them.

## Limitations

Our present study has several limitations. The single most important of these is the timeframe of the search, which occurred immediately preceding the COVID-19 pandemic. We recognize that the pandemic triggered a surge of interest in remote monitoring and wearable technologies [29,30], and their exclusion paints an incomplete picture of the full breadth of devices and innovations available for use in the diagnosis and management of coronavirus-like communicable diseases. Nevertheless, we hope that the timing of the article search allows the reader to understand what the ecosystem of mHealth independent of COVID-19 looked like, as the pandemic was responsible for significant resource-shifting away from pre-existing infectious diseases of substantial importance in LMICs, particularly neglected tropical diseases [31–33]. Future systematic reviews of this sphere taking into account mobile innovations for SARS-CoV-2-related disease will be instrumental in characterizing the full scope of the technological armamentarium available in the ongoing post-COVID-19 world.

Next, from a search strategy perspective, we employed a restrictive set of inclusion criteria, which excluded patient-facing devices and apps which digitized communication, algorithms/guidelines, and clinical calculators. Such technologies may have important impacts on health outcomes in resource-poor settings but were outside the scope of our review. Thus, their notable contribution to the overall ecosystem of mHealth interventions for communicable diseases in LMICs is not available here for context. In fact, the search strategy did not include specific terms alluding to these knowledge-based algorithms such as Clinical Decision Support System (CDSS), or Clinical Decision Support Algorithms (CDSA) or machine learning / artificial intelligence technologies. The decision to exclude these specific terms was made to prioritize mobile health innovations with clear applicability and practical usability in LMICs. Nevertheless, we acknowledge that this approach may have resulted in the exclusion of some relevant articles that specifically focused on these technologies and their interface with in the realm of mHealth.

Lastly, from a methodological standpoint, the heterogeneity of the included studies regarding their results and methodological approaches precluded us from performing a meta-analysis and systematic assessment of study quality, necessitating a qualitative grading system instead. Additionally, we did not conduct a duplicate database search, and while our single investigator system ensured consistency in the screening process, this approach could have resulted in the possibility of rejecting relevant reports. This issue is particularly highlighted by the fact that one of our study inclusion criteria was that the mHealth technology must represent an innovation (and not reproduce existing guidelines)—in this regard, there could have been subjectivity introduced into the screening process that may affect the reproducibility of our work.

## Conclusions

This systematic review found that there are only a small number of mHealth technologies that constitute novel methods of screening, diagnosing, or monitoring communicable diseases of public health importance in LMICs. Randomized trials and evaluations with large sample sizes of these technologies are still lacking, as are applications meant to monitor diseases. Additionally, most identified products require accessories or peripheral devices, and a majority rely on operating systems not common in LMICs, thus likely precluding more widespread clinical use

in these settings. Future studies should examine the impact of COVID-19 on the ecosystem of these devices as well, given rapid, sweeping changes in mHealth catalyzed by the pandemic.

## Supporting information

**S1 PRISMA Checklist. PRISMA checklist.**
(DOCX)

**S1 Table. This table presents the full literature search strategy, listing all of the search terms used for each database queried.**
(DOCX)

**S2 Table. This table lists our analytic themes, as well as the categories, subcategories, and definitions that accompany them.**
(DOCX)

**S3 Table. This table presents the full list of studies identified by our search and screen, which were ultimately analyzed in our synthesis.**
(DOCX)

## Acknowledgments

Innovation schematics in Fig 2 are reproduced from the cited publications and remain the property of their respective authors/publishers. Attribution links are provided for the image sources and licenses.

## Author Contributions

**Conceptualization:** Pascal Geldsetzer, Sergio Flores, Blanca Flores, Abu Bakarr Rogers.

**Data curation:** Pascal Geldsetzer, Sergio Flores, Blanca Flores, Abu Bakarr Rogers.

**Formal analysis:** Pascal Geldsetzer, Sergio Flores, Andrew Y. Chang.

**Funding acquisition:** Pascal Geldsetzer.

**Investigation:** Pascal Geldsetzer, Sergio Flores, Abu Bakarr Rogers, Andrew Y. Chang.

**Methodology:** Pascal Geldsetzer, Sergio Flores, Blanca Flores, Abu Bakarr Rogers, Andrew Y. Chang.

**Project administration:** Pascal Geldsetzer, Sergio Flores, Blanca Flores, Andrew Y. Chang.

**Resources:** Pascal Geldsetzer.

**Supervision:** Pascal Geldsetzer, Andrew Y. Chang.

**Writing – original draft:** Sergio Flores.

**Writing – review & editing:** Pascal Geldsetzer, Sergio Flores, Andrew Y. Chang.

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
