## [Decision Letter · Decision Letter 0]

20 Apr 2023

PDIG-D-22-00325

Healthcare provider-targeted mobile applications to diagnose, screen, or monitor communicable diseases of public health importance in low- and middle-income countries: a systematic review

PLOS Digital Health

Dear Dr. Chang,

Thank you for submitting your manuscript to PLOS Digital Health. After careful consideration, we feel that it has merit but does not fully meet PLOS Digital Health's publication criteria as it currently stands. Therefore, we invite you to submit a revised version of the manuscript that addresses the points raised during the review process.

Please submit your revised manuscript within 60 days Jun 19 2023 11:59PM. If you will need more time than this to complete your revisions, please reply to this message or contact the journal office at digitalhealth@plos.org. Please include the following items when submitting your revised manuscript:

We look forward to receiving your revised manuscript.

Kind regards,

Ana Luísa Neves

Academic Editor

PLOS Digital Health

Journal Requirements:

1. Please send a completed 'Competing Interests' statement, including any COIs declared by your co-authors. If you have no competing interests to declare, please state "The authors have declared that no competing interests exist". Otherwise please declare all competing interests beginning with the statement "I have read the journal's policy and the authors of this manuscript have the following competing interests:"

3. Please provide separate figure files in .tif or .eps format only and remove any figures embedded in your manuscript file. Please also ensure that all files are under our size limit of 10MB.

Editor's comments:

The search was performed in October 2019, nearly 4 years from the date of expected publication. Would be recommended to update the results - if this is not possible this should be clearly outlined in the limitations.

Reviewers' comments:

Reviewer #1: Amazing contribución as a physician in a country in developpment I consider accurate a review like the one you have developped. No other commentary, statically accurate, well designed and a very useful contribution

Reviewer #2: Thank you very much for inviting me to review this systematic review. 

The authors performed a literature search for mHealth technologies aimed at use in low- and middle-income countries. Such technologies must be mobile-based and aimed at use by healthcare professionals. It is interesting to note that the majority of studies included in the review (48%) were performed in the United States or Europe.

My main comment relates to scope of the inclusion and exclusion criteria employed in the study which in turn limits the impact or the learnings which can be taken away from this systematic review. As a technical piece the review fulfils the criteria set out by the authors but I would argue that the utility of the review and interest is limited. 

- The review excluded any interventions which were not considered an “innovation” in mHealth, but at the same time many of the diagnostics means required additional equipment or reagents to operate. For example, Li et al., 2018’s paper included presented an aptameric assay for Mtb which is read using a smartphone. Although fulfilling the authors’ criteria I would argue that this study represents the development of a point of care test which incidentally makes use of a mobile phone to display its results, rather than being a mHealth innovation in itself which I understand is the aim of the review. 

- The authors specifically excluded interventions which were “merely” digitalised paper-based tools, or “simply” facilitated communication. Whilst this is important in their restricted definition of the systematic review intentions, the language used does bely the utility of truly mHealth innovations which aim to make use of simple interventions to translate to actual utility. 

- The inclusion date of the studies included ended in 2019 which limits the relevance of this review, given that COVID-19 was a significant catalyst to the development of smartphone and wearable-based innovations in mHealth. As a result there were no wearables studies (Armband/Smartwatch) included e.g. https://doi.org/10.1016/S2589-7500(19)30222-5 and https://doi.org/10.1038/s41591-020-1123-x which is a shame. One could perform an update of the review to present date specifically excluding COVID-19 as a condition for example. 

- The majority of studies were of product descriptions only which restricts the authors’ ability to perform a meta-analysis or examine impact. 

The difficulty in maintaining this a narrow scope of review is appreciated but given the paucity of relevant studies which fit into the authors’ restrictive criteria, I would like to see this broadened to include more studies to current date, or a broader range of conditions which make use of innovative mHealth technologies which are extremely relevant to LMICs e.g. retinal screening and other non-communicable diseases interventions.

Reviewer #3: The authors performed a systematic review on mobile healthcare provider-targeted mobile applications to diagnose, screen or monitor communicable diseases in LMICs. With their search terms they identified 33 studies meeting their inclusion criteria from 13262 identified by the first screen. Interestingly, almost all mobile apps were of diagnostic nature and almost none was used for monitoring the diseases. 

In general, it is an interesting approach to perform such an analysis. The work is well done and the paper well written.

What is missing is to give the reader a clearer idea, what these apps actually can do, respectively should do. It would be of great help, if the authors could maybe categorize the different apps into groups but then provide clear examples what was really done, respectively what this apps can do (maybe do a "visual" table with screen shots etc. of what these apps really provide. it is not clear what an app can do regarding a serological diagnosis, respectively how e.g. parasites are visualized??

Line 242: here the word "save" does not make sense. Sentence should be rewritten.

Reviewer #4: The authors perform a systematic review on a subtype of mobile technologies to support healthcare workers, outlining studies and tools that differ from knowledge based clinical decision support algorithms. This is indeed a category of digital health devices that have been less described in previous reviews, and so a helpful contribution to better understanding the landscape of mobile health digital tools for healthcare workers. There are however some significant deviations to the protocol that have not been described, and clarity is needed on some of the approaches. The search was performed nearly 3.5 years ago limiting its relevance in a field that is constantly changing. I would propose a major revision to the manuscript before considering publication.

Minor concerns/comments:

1. In order for the reader to understand the appropriateness of the search strategy it would be necessary to understand the eligibility criteria. Would consider moving the eligibility criteria before the search strategy. This also aligns with the order proposed by PRISMA.

2. The authors present the full search strategy in the appendix, but would be helpful to understand the simplified search strategy in the main text. i.e. Combining the following search terms “mobile/tablet” and “application/software” and “diagnostics/monitoring”

3. The authors clearly state that the database search was not conducted in duplicate which may result in the possibility of rejecting relevant reports. Nonetheless it can be justified if the selection process is quite clear-cut.

a. Please address this limitation in the discussion.

b. What is unclear is if the screening was performed by one person or multiple. If multiple people, describe limitations this could have resulted in this process.

4. Line 176: I would remove “and the extremely rapid turnover in the science surrounding the disease and its many novel variants”. The search was performed in October 2019, as such it is the only reason why COVID-19 was not included.

5. Figure 1: 

a. 48 manuscripts were excluded because they describe “current technologies” This is not clear and I am unable to make the link with the inclusion/exclusion criteria. Can the authors please clarify?

6. The search strategy does not include names of digital health tools that are typically associated to the technologies being looked for: these include “mHealth”, “Clinical Decision Support System”, “CDSS”, “Clinical Decision Support Algorithm”, “CDSA”, “eHealth”. Would suggest looking at other mHealth systematic reviews for established examples. Can the authors explain why these were not included in the search strategy and comment on this in the limitations. Was there a reason for not also including approaches to the search (ex. Machine learning, artificial intelligence)?

Major:

1. Inclusion criteria:

1.1. The following inclusion criteria were not pre-specified in the protocol: Technology must target healthcare professionals (line 160) and Technology must represent an innovation (line 165)

1.1.1. I don’t consider the first modification to be a significant deviation to the protocol as it is in some way implied in the protocol, however the second modification is a much bigger modification to the protocol. The authors should clarify when this inclusion criteria was added (before or after the start of the search, before or during the screening process), and why this inclusion criteria was added. If added during the search process, please clarify if the screening process was restarted given the change in search strategy.

1.1.2. Furthermore the inclusion criteria definition of “must represent an innovation” “reproduce existing guidelines” is not straightforward and thus vulnerable to personal interpretation. This would hinder reproduction of such a systematic review. In reference to the minor issue highlighted in point 3, this would be a good justification for using at least two people to screen research articles. The search should either be done once again by a second person, or clearly outlined as a limitation in the discussion.

2.1.3. Line 228 the inclusion/exclusion criteria previously described in line 165 is different. Would suggest to use the most detailed description in the methods.

2.1.4. Based on the inclusion criteria, I am unclear why numerous electronic clinical decision support systems (IEDA/REC, eIMCI, Medsinc, ALMANACH, ePOCT, etc) were excluded. I assume it may be due to the “not reproduce existing guidelines” criteria. Of note IEDA has a respiratory count aid similar to that as mPneumonia (included in the authors’ review), however I am unsure this is described in the publications. Many of the other tools mentioned calculates drug doses, and z-scores, would this not meet inclusion criteria? Many were also not just digitization of paper guidelines, as many included significant changes to the clinical algorithms. While I think sensible that the authors concentrate on non-knowledge based mHealth tools, this is not quite clear in the inclusion criteria. Would consider clarifying this inclusion criteria by including the concept of “knowledge based algorithm” as described by Papadopoulos et al.: (https://link.springer.com/article/10.1007/s12553-022-00672-9) This of course would only to help reproduce such a review by other research groups.

---

## [Decision Letter · Decision Letter 1]

11 Aug 2023

Healthcare provider-targeted mobile applications to diagnose, screen, or monitor communicable diseases of public health importance in low- and middle-income countries: a systematic review

PDIG-D-22-00325R1

Dear Dr. Chang,

We are pleased to inform you that your manuscript 'Healthcare provider-targeted mobile applications to diagnose, screen, or monitor communicable diseases of public health importance in low- and middle-income countries: a systematic review' has been provisionally accepted for publication in PLOS Digital Health.

Best regards,

Ana Luísa Neves

Academic Editor

PLOS Digital Health

Reviewer Comments (if any, and for reference):

Reviewer's Responses to Questions

**Comments to the Author**

1. If the authors have adequately addressed your comments raised in a previous round of review and you feel that this manuscript is now acceptable for publication, you may indicate that here to bypass the “Comments to the Author” section, enter your conflict of interest statement in the “Confidential to Editor” section, and submit your "Accept" recommendation.

Reviewer #3: All comments have been addressed

Reviewer #4: All comments have been addressed

2. Does this manuscript meet PLOS Digital Health’s publication criteria? Is the manuscript technically sound, and do the data support the conclusions? The manuscript must describe methodologically and ethically rigorous research with conclusions that are appropriately drawn based on the data presented.

Reviewer #3: Yes

Reviewer #4: Yes

3. Has the statistical analysis been performed appropriately and rigorously?

Reviewer #3: N/A

Reviewer #4: N/A

4. Have the authors made all data underlying the findings in their manuscript fully available (please refer to the Data Availability Statement at the start of the manuscript PDF file)?

Reviewer #3: Yes

Reviewer #4: Yes

5. Is the manuscript presented in an intelligible fashion and written in standard English?

Reviewer #3: Yes

Reviewer #4: Yes

6. Review Comments to the Author

Reviewer #3: The authors answered the questions of the reviewers at best.

Reviewer #4: I congratulate the authors again on this well written and important systematic review. In my opinion the revisions address all the concerns of the reviewers, and was clearly written.

Although not mandatory, one point to consider mentioning in regards to the scope of the paper is the fact that while "innovative methods" may be important to improving care, these "innovative methods" often translate to less "explainability", and worse "black box algorithms". Such approaches sometimes are indeed more accurate than more explainable models, but may hinder clinician understanding of the tools and decisions, which in turn may impact clinical and patient autonomy, continued learning, and fostering trust for a good physician-patient relationship.

7. PLOS authors have the option to publish the peer review history of their article (what does this mean?). If published, this will include your full peer review and any attached files.

**Do you want your identity to be public for this peer review?** For information about this choice, including consent withdrawal, please see our Privacy Policy.

Reviewer #3: No

Reviewer #4: **Yes: **Rainer Tan
